# Whole-Body Pharmacokinetics and Physiologically Based Pharmacokinetic Model for Monomethyl Auristatin E (MMAE)

**DOI:** 10.3390/jcm10061332

**Published:** 2021-03-23

**Authors:** Hsuan Ping Chang, Yuen Kiu Cheung, Dhaval K. Shah

**Affiliations:** Department of Pharmaceutical Sciences, School of Pharmacy and Pharmaceutical Sciences, The State University of New York at Buffalo, Buffalo, NY 14214, USA; hsuanpin@buffalo.edu (H.P.C.); yuenkiuc@buffalo.edu (Y.K.C.)

**Keywords:** antibody–drug conjugate (ADC), monomethyl auristatin E (MMAE), biodistribution, tissue pharmacokinetics (PK), physiological-based pharmacokinetic (PBPK) model

## Abstract

Monomethyl auristatin E (MMAE) is one of the most commonly used payloads for developing antibody–drug conjugates (ADC). However, limited studies have comprehensively evaluated the whole-body disposition of MMAE. Consequently, here, we have investigated the whole-body pharmacokinetics (PK) of MMAE in tumor-bearing mice. We show that while MMAE is rapidly eliminated from the plasma, it shows prolonged and extensive distribution in tissues, blood cells, and tumor. Highly perfused tissues (e.g., lung, kidney, heart, liver, and spleen) demonstrated tissue-to-plasma area under the concentration curve (AUC) ratios > 20, and poorly perfused tissues (e.g., fat, pancreas, skin, bone, and muscle) had ratios from 1.3 to 2.4. MMAE distribution was limited in the brain, and tumor had 8-fold higher exposure than plasma. A physiological-based pharmacokinetic (PBPK) model was developed to characterize the whole-body PK of MMAE, which accounted for perfusion/permeability-limited transfer of drug in the tissue, blood cell distribution of the drug, tissue/tumor retention of the drug, and plasma protein binding. The model was able to characterize the PK of MMAE in plasma, tissues, and tumor simultaneously, and model parameters were estimated with good precision. The MMAE PBPK model presented here can facilitate the development of a platform PBPK model for MMAE containing ADCs and help with their preclinical-to-clinical translation and clinical dose optimization.

## 1. Introduction

Antibody–drug conjugates (ADCs) have become a promising class of drug molecules for the treatment of cancer. There have been nine ADCs approved by the United States Food and Drug Administration (FDA), and more than 80 ADCs are in clinical trials [1]. Monomethyl auristatin E (MMAE) is one of the most commonly used payloads to make ADCs, and Adcetris^®^ (brentuximab vedotin), Padcev^®^ (enfortumab vedotin-ejfv), and Polivy^®^ (polatuzumab vedotin-piiq) are three clinically approved ADCs that contain MMAE [2]. After ADC internalization, the released MMAE in the tumor cells can enter surrounding cells and cause bystander killing [3]. This advantage of MMAE leads to an efficient killing of tumor cells. However, it can also cause toxicity to healthy cells. While MMAE-conjugated ADCs are efficacious, hematological adverse reactions such as neutropenia (≈21%) and thrombocytopenia (≈10%) are consistently reported [4,5]. Moreover, peripheral neuropathy (≈44%) is also a predominant adverse effect in the clinic (≈44%) [5,6]. It is generally believed that these toxicities stem from the pharmacological effects of the payload. However, limited studies have investigated whole-body pharmacokinetics (PK) of MMAE. While it is impossible to conduct such studies in humans due to the inherent toxicity of MMAE, preclinical studies that investigate whole-body PK of MMAE are required to better understand the disposition of this molecule and to facilitate preclinical-to-clinical translation of exposure–response relationships developed for MMAE-conjugated ADCs.

Systemic PK properties of MMAE, such as the elimination pathway, have been reported based on PK studies of brentuximab vedotin [5,7] and MMAE [8]. The elimination of MMAE is predominantly through the CYP3A4/5-mediated metabolic pathway and biliary/fecal excretion, with limited renal excretion [4,5,9]. The mechanism of action of MMAE is known, as it inhibits cell division by binding to tubulin dimers and disrupting the microtubule network [4]. The binding of MMAE to its cellular target, tubulin, can lead to extensive and prolonged drug exposure into tissues, and thus plasma exposure alone may not represent tissue exposure. Tissue distribution of MMAE has been studied in rats in a radioactivity study [8] and a quantitative whole-body autoradiography study [7]. Yip et al. [8] studied the disposition and mass balance of MMAE following intravenous administration of 3H-labeled MMAE at a dose of 200 µg/kg. The measurement of tissue radioactivity showed fast distribution of MMAE to highly perfused organs such as the liver, lungs, and kidneys. Pastuskovas et al. [7] characterized the tissue distribution of Herceptin-vc-[^14^C]MMAE following an intravenous dose (11 mg/kg) of the radiolabeled ADC in rats, which was analyzed by quantitative whole-body autoradiography. The study characterized that Herceptin-vc-[^14^C]MMAE distributes to highly perfused organs, and the majority of blood radioactivity represented radiolabeled ADC with low levels of free drug. However, both studies only described the pattern of MMAE tissue distribution, and detailed and quantitative information about the concentration profiles and PK parameters (i.e., area under the concentration curve, AUC) in tissues were not reported. Therefore, there is a need for more quantitatively thorough studies that investigate whole-body distribution of MMAE.

There have also been studies that use mathematical modeling to characterize the systemic PK of MMAE-conjugated ADCs. Chen et al. built a minimal physiologically-based pharmacokinetic (PBPK) model to predict drug–drug interactions for MMAE-conjugated ADCs in humans [9]. The population PK modeling approach has also been employed to investigate the PK variability of brentuximab vedotin in adults [10] and pediatrics [11]. However, these models did not serve the purpose of predicting the whole-body distribution of MMAE. On the other hand, the vc-cleavable-linker is commonly used for MMAE-conjugated ADCs, which is different from the non-cleavable linker used in Ado-trastuzumab emtansine (T-DM1) [12]. Once MMAE-conjugated ADC is internalized, the vc-linker can be cleaved by protease in the lysosome, and free MMAE is released to the systemic circulation. As such, the PK of unconjugated MMAE behaves the same as MMAE administrated in the free form. Therefore, the development of a PBPK model for free MMAE can be useful to characterize the PK of unconjugated MMAE following the administration of MMAE-conjugated ADCs, and for the development of exposure–response relationships to predict the toxicity of these ADCs.

In this manuscript, we have investigated the whole-body biodistribution of MMAE in tumor-bearing mice, and we have developed a PBPK model to characterize the plasma, tissues, and tumor PK of MMAE. The PBPK model for MMAE developed here can further serve as a framework for the development of a platform PBPK model for MMAE-conjugated ADCs [13].

## 2. Materials and Methods

### 2.1. Chemical and Reagents

MMAE (purity > 98%) and D8-MMAE (internal standard (IS), purity > 99%) were purchased from MedChem Express (Monmouth Junction, NJ, USA). Acetonitrile, dimethyl sulfoxide, radioimmunoprecipitation assay (RIPA) buffer, and 1X Halt™ protease inhibitor were purchased from Fisher Scientific (Watham, MA, USA). Formic acid was purchased from Sigma Aldrich (St. Louis, MO, USA). Ultrapure water purified by Barnstead Nanopure Diamond system was used in this study (Dubuque, IA, USA)

### 2.2. Development of Xenograft Mouse Model

The breast cancer cell line MDA-MB-468 (ATCC^®^ HTB-132), purchased from American Type Tissue Culture (Manassas, VA, USA), was used to develop the xenograft tumors. Cells were grown in the RPMI1640 medium (ATCC^®^ 302001™) supplemented with heat-inactivated 10% *v*/*w* fetal bovine serum (Gibco/Thermo Fisher Scientific, Grand Island, NY, USA) and 10 μg/mL of gentamycin (Sigma-Aldrich, St. Louis, MO, USA). Cells were cultured in a humidified incubator maintained with 5% carbon dioxide at 37 °C.

Male athymic nude mice were purchased at the age of 6 weeks from Charles River (Wilmington, NC, USA). After acclimation to the new conditions for two weeks, mice were subcutaneously injected with MDA-MB-468 (about 10 million tumor cells) into the right dorsal flank. The in vivo study adhered to the Principles of Laboratory Animal Care (National Institutes of Health publication 85–23, revised 1985) and were approved by the University at Buffalo Institutional Animal Care and Use Committee (IACUC# PHC29035Y).

### 2.3. Biodistribution Study

A total of 18 mice (6 weeks old, 26–35 g) bearing MDA-MB-468 xenografts were used for the biodistribution studies. Then, 0.1 mg/kg MMAE was injected into the mice via the penile vein, and terminal samples were collected at 5 min and 1, 6, 12, 24, and 168 h. Three mice were sacrificed at each time point. Whole blood, tissue, and tumor were harvested. The collected tissues included the heart, liver, lung, spleen, pancreas, kidney, skin, bone, muscle, fat, and brain. Whole blood samples in ethylenediaminetetraacetic acid (EDTA) pre-coated tubes were centrifuged at 2000× *g* for 20 min at 4 °C, and plasma was collected and stored at −20 °C for further analysis. Harvested tissue samples were blotted dry and immediately frozen in liquid nitrogen and stored at −80 °C until homogenization.

### 2.4. Sample Preparation and LC-MS/MS Quantification of MMAE

A detailed tissue homogenization procedure has been reported previously [14]. Briefly, different volumes of RIPA buffer containing protease inhibitor were added into the weighted tissue samples to obtain different dilution factors in each tissue. The dilution factor was 5 for heart, liver, lung, spleen, pancreas, kidney, skin, and fat, 8 for bone and muscle, and 4 for brain samples. Tissue samples were homogenized using a BeadBug™ microtube homogenizer (Benchmark, USA) at the maximum speed for 15 s followed by a 30-s ice cool down, and repeated three times. Blood samples without dilution were directly homogenized for 30 s at maximum speed and were treated the same way as tissue samples. 

Then, 500 µL acetonitrile containing 0.1% formic acid was added into 100 µL of plasma, tissue, or tumor homogenate samples and then spiked with 20 µL of IS solution (D8-MMAE 150 ng/mL). After vortexing and centrifugation at 15,000× *g* for 15 min at 4 °C, the supernatants were transferred to glass tubes and dried under nitrogen flow at 32 °C. The dried residuals were reconstituted with 60 µL of acetonitrile/water (95:5 *v*/*v*) containing 0.1% formic acid, gently vortexed, and immediately transferred into HPLC vials. 

Standards and quality control samples (QCs) were prepared for plasma and each tissue matrix. Then, 250 µL of the control plasma or matrices were spiked with 20 µL of IS solution (D8-MMAE 150 ng/mL) and 10 µL of MMAE stock solution diluted with acetonitrile, and then, 500 µL of acetonitrile containing 0.1% formic acid was added. The final concentrations of standard were 0.5, 1, 10, 100, 200, and 500 ng/mL. QCs were prepared similarly for the final concentrations of 5, 50, and 250 ng/µL.

A Waters Acquity LC-MS/MS system was used with electrospray interphase and triple quadrupole mass spectrometer. An ACQUITY UPLC BEH Amide column (Waters, Milford, MA, USA) was used with 5 mM ammonium formate and 0.1% formic acid as the aqueous phase, and 0.1% formic acid and 1 mM ammonium formate were used as the organic phase. The gradient flow rate was 0.25 mL/min. The lower limit of quantification of MMAE was 0.2 ng/mL (or ng/g) for plasma, blood, tissues, and tumor.

### 2.5. Data Analysis

Noncompartmental analysis (NCA) was conducted for plasma PK data. AUC computed from time 0 to the last observed concentration time (*AUC*_0−*t*_) was calculated using the linear/log trapezoidal method in WinNonlin (version 8.1, Pharsight, St. Louis, MO, USA). Blood to plasma ratios ($K_{P,BC}$) and tissue partition coefficients ($K_{P,i}$) were calculated using Equation (1), using *AUC*_0−*t*_ values derived from the observed data.
(1)$$K_{P,BC}\;or\;K_{P,i}=\frac{AUC_{i,0-t}}{AUC_{p,0-t}},$$

Above, $AUC_{i,0-t}$ is $AUC_{0-t}$ in the blood cell, tumor, or tissue *i*; $AUC_{p,0-t}$ is $AUC_{0-t}$ in plasma. It was assumed that MMAE resided mainly within the cellular compartment of tissues, and hence, the $K_{P,i}$ value for each tissue was adjusted by dividing $K_{P,i}$ with the fractional cellular space volume of each tissue for PBPK modeling purposes.

### 2.6. PBPK Model Development

#### 2.6.1. Model Structure

Figure 1a shows the proposed PBPK model structure for MMAE. The PBPK model included 16 tissues and a tumor compartment, and all of them were connected via blood flow and arranged anatomically. The tissue compartments included blood, lung, heart, kidney, muscle, skin, liver, brain, adipose, thymus, bone, small intestine, large intestine, spleen, pancreas, and other (i.e., carcass). All the other tissues, except those mentioned, were lumped into the ‘other’ compartment. The arterial blood to each organ was delivered by the efferent blood supplied from the lung, which perfused to each organ and then converged into the blood compartment, which represents the venous pool. Venous blood returned from the small intestine, large intestine, spleen, and pancreas were delivered to the liver. For the intestines, spleen, and pancreas, blood was delivered to the liver via the hepatic portal vein and mixed with liver artery blood after leaving the tissues. The delivery of the blood from the blood compartment to the lung completed the circulation of the flow. Based on in vitro, preclinical, and clinical data, the elimination of MMAE was assumed to be predominantly through the CYP3A4/5-mediated metabolic pathway and biliary/fecal excretion [4,5,9]. Limited renal excretion (<10%) of MMAE has been reported [4,5,7]. Therefore, we assumed that MMAE was eliminated solely via hepatic clearance (CL) from the liver interstitial space. 

Each tissue compartment was further divided into the vascular, endothelial cell, interstitial, and cellular sub-compartments, and vascular space was divided into plasma and blood cells, as shown in Figure 1b. We aimed to make the sub-compartment division the same as our previously published platform antibody PBPK model, and thus, one can easily connect MMAE and an antibody PBPK model to build an ADC PBPK model [13]. The rapid distribution of MMAE between plasma, endothelial cells, and interstitial space was assumed, and thus, the distribution rate between these compartments was set as a value 1000 times higher than the value of blood flow in each tissue. The accumulation profiles of the drug in tissues suggested that MMAE distribution between plasma and the blood cells, and between interstitial and cellular spaces, were permeability-limited, and thus permeability coefficients ($PS_{i}$) that represented passive diffusion in each tissue were employed. The partitioning of MMAE to the blood cells or cellular space was characterized using the $K_{P}$ values. The calculated $K_{P}$ values were adjusted for the ratios of cellular space volume to total tissue volume by dividing $K_{P}$ values with the ratio in each tissue. The plasma protein binding of MMAE has been reported to be 17.1–28.5% in mice and monkey [4], and hence, we assumed 20%, which means that 80% of MMAE is unbound in the plasma ($f_{u,p}$). Only free MMAE was assumed to diffuse through the cellular membrane, and thus, the fraction of unbound drug in blood cells or tissue cellular space ($f_{u,t}$) was calculated using Equation (2) [15].
(2)$$f_{u,t}=\frac{f_{u,p}}{K_{P,i}},$$

Characterization and prediction of MMAE concentration at the site of action (i.e., tumor) is essential to establish a reliable exposure–efficacy relationship. Therefore, a cell-level tumor disposition model was incorporated in the MMAE PBPK model [16]. In the tumor model, as shown in Figure 1c, MMAE was allowed to move between the plasma and tumor extracellular space by the vascular exchange (extravasation) and surface exchange (diffusion) pathway, which was defined by the coefficient of permeability (P) and diffusion (D), respectively. Both pathways also depended on the vascular density and tumor size (Equation (20)). MMAE in the tumor extracellular space can influx into the tumor cell and bind to the tubulin. The tubulin-bound MMAE can dissociate from the target and efflux from tumor intracellular space into the extracellular space.

#### 2.6.2. Model Equations

Equations (3) and (4) describe blood cell and plasma concentrations of MMAE, respectively. Equations that described MMAE concentration in the liver (Equations (5)–(9)), lung (Equations (10)–(14)), a typical tissue (Equations (15)–(19)), and tumor (Equations (20)–(23)) are provided below. In these equations, $Q_{plasma}^{i}$ and $Q_{BC}^{lung}$ are plasma flow and blood cell flow to the tissue *i.*
$V_{plasma}$ and $V_{BC}$ are volumes of central plasma and blood cell compartments. $V_{plasma}^{i}$, $V_{BC}^{i}$, $V_{endo}^{i}$, $V_{IS}^{i}$, and $V_{cellular}^{i}$ are volumes of vascular, blood cell, endosomal, interstitial, and cellular compartments for tissue *i*. $C_{plasma}$ and $C_{BC}$ are MMAE concentration in systemic plasma and blood cell compartments. $C_{plasma}^{i}$, $C_{BC}^{i}$, $C_{endo}^{i}$, $C_{IS}^{i}$, and $C_{cellular}^{i}$ are MMAE concentration in vascular, blood cell, endosomal, interstitial, and cellular compartments of tissue *i.*
$PS_{BC}$ is the permeability coefficient for the blood cell sub-compartment. A glossary of tumor-associated parameters is provided in Table 1.

Blood compartment


Plasma
(3)$$V_{plasma}\times \frac{dC_{plasma}}{dt}=Q_{plasma}^{kidney}\times C_{plasma}^{kidney}+Q_{plasma}^{heart}\times C_{plasma}^{heart}+Q_{plasma}^{brain}\times C_{plasma}^{brain}+Q_{plasma}^{skin}\times C_{plasma}^{skin}+\;Q_{plasma}^{muscle}\times C_{plasma}^{muscle}+Q_{plasma}^{bone}\times C_{plasma}^{bone}+Q_{plasma}^{fat}\times C_{plasma}^{fat}+Q_{plasma}^{thymus}\times C_{plasma}^{thymus}+(Q_{plasma}^{liver}+Q_{plasma}^{spleen}+\;Q_{plasma}^{muscle}+Q_{plasma}^{SI}+Q_{plasma}^{LI})\times C_{plasma}^{liver}+Q_{plasma}^{carcass}\times C_{plasma}^{carcass}-Q_{plasma}^{lung}\times C_{plasma}+PS_{BC}\times C_{BC}\times \frac{f_{u,p}}{K_{P,BC}}-\;PS_{BC}\times C_{plasma}\times f_{u,p}-\left(\frac{2\times P\times R_{cap}}{R_{krough}^{2}}\right)\times (C_{plasma}-\frac{C_{free,extra}^{tumor}}{\varepsilon })\times V_{tumor}-\left(\frac{6\times D}{R_{tumor}^{2}}\right)\times \left(C_{plasma}-\frac{C_{free,extra}^{tumor}}{\varepsilon }\right)\times \;V_{tumor},$$Blood cells
(4)$$V_{BC}\times \frac{dC_{BC}}{dt}=Q_{BC}^{kidney}\times C_{BC}^{kidney}+Q_{BC}^{heart}\times C_{BC}^{heart}+Q_{BC}^{brain}\times C_{BC}^{brain}+Q_{BC}^{skin}\times C_{BC}^{skin}+Q_{BC}^{muscle}\times C_{BC}^{muscle}+\;Q_{BC}^{bone}\times C_{BC}^{bone}+Q_{BC}^{fat}\times C_{BC}^{fat}+Q_{BC}^{thymus}\times C_{BC}^{thymus}+\left(Q_{BC}^{liver}+Q_{BC}^{spleen}+Q_{BC}^{pancreas}+Q_{BC}^{SI}+Q_{BC}^{LI}\right)\times C_{BC}^{liver}+\;Q_{BC}^{carcass}\times C_{BC}^{carcass}-Q_{BC}^{lung}\times C_{BC}+PS_{BC}\times C_{plasma}\times f_{u,p}-PS_{BC}\times C_{BC}\times \frac{f_{u,p}}{K_{P,BC},}$$Liver compartmentPlasma
(5)$$V_{plasma}^{liver}\times \frac{dC_{plasma}^{liver}}{dt}=Q_{plasma}^{liver}\times C_{plasma}^{lung}+Q_{plasma}^{spleen}\times C_{plasma}^{spleen}+Q_{plasma}^{pancreas}\times C_{plasma}^{pancreas}+Q_{plasma}^{SI}\times C_{plasma}^{SI}+\;Q_{plasma}^{LI}\times C_{plasma}^{LI}-\left(Q_{plasma}^{liver}+Q_{plasma}^{spleen}+Q_{plasma}^{pancreas}+Q_{plasma}^{SI}+Q_{plasma}^{LI}\right)\times C_{plasma}^{liver}+PS_{BC}\times C_{BC}^{liver}\times \;\frac{f_{u,p}}{K_{P,BC}}-PS_{BC}\times C_{plasma}^{liver}\times f_{u,p}-Q_{plasma}^{liver}\times G\times C_{plasma}^{liver}\times f_{u,p}+Q_{plasma}^{liver}\times G\times C_{endo}^{liver},$$Blood cells
(6)$$V_{BC}^{liver}\times \frac{dC_{BC}^{liver}}{dt}=Q_{BC}^{liver}\times C_{BC}^{lung}+Q_{BC}^{spleen}\times C_{BC}^{spleen}+Q_{BC}^{pancreas}\times C_{BC}^{pancreas}+Q_{BC}^{SI}\times C_{BC}^{SI}+Q_{BC}^{LI}\times C_{BC}^{LI}-\;\left(Q_{BC}^{liver}+Q_{BC}^{spleen}+Q_{BC}^{pancreas}+Q_{BC}^{SI}+Q_{BC}^{LI}\right)\times C_{BC}^{liver}+PS_{BC}\times C_{plasma}^{liver}\times f_{u,p}-PS_{BC}\times C_{BC}^{liver}\times \frac{f_{u,p}}{K_{P,BC}},$$Endothelial cell
(7)$$V_{endo}^{liver}\times \frac{dC_{endo}^{liver}}{dt}=Q_{plasma}^{liver}\times G\times f_{u,p}\times C_{plasma}^{liver}-2\times Q_{plasma}^{liver}\times G\times C_{endo}^{liver}+Q_{plasma}^{liver}\times G\times C_{IS}^{liver},$$Interstitial space
(8)$$V_{IS}^{liver}\times \frac{dC_{IS}^{liver}}{dt}=Q_{plasma}^{liver}\times G\times C_{endo}^{liver}-Q_{plasma}^{liver}\times G\times C_{IS}^{liver}-PS_{liver}\times C_{IS}^{liver}+PS_{liver}\times C_{cellular}^{liver}\times \;\frac{f_{u,p}}{K_{P,liver}}-CL_{int}\times C_{IS}^{liver},$$Cellular space
(9)$$V_{cellular}^{liver}\times \frac{dC_{cellular}^{liver}}{dt}=PS_{liver}\times C_{IS}^{liver}-PS_{liver}\times C_{cellular}^{liver}\times \frac{f_{u,p}}{K_{P,liver}},$$Lung CompartmentPlasma
(10)$$V_{plasma}^{lung}\times \frac{dC_{plasma}^{lung}}{dt}=Q_{plasma}^{lung}\times C_{plasma}-(Q_{plasma}^{heart}+Q_{plasma}^{kidney}+Q_{plasma}^{brain}+Q_{plasma}^{muscle}+Q_{plasma}^{bone}+Q_{plasma}^{thymus}+\;Q_{plasma}^{skin}+Q_{plasma}^{fat}+Q_{plasma}^{liver}+Q_{plasma}^{SI}+Q_{plasma}^{LI}+Q_{plasma}^{spleen}+Q_{plasma}^{pancreas}+Q_{plasma}^{other})\times C_{plasma}^{lung}+PS_{BC}\times \;C_{BC}^{lung}\times \frac{f_{u,p}}{K_{P,BC}}-PS_{BC}\times C_{plasma}^{lung}\times f_{u,p}-Q_{plasma}^{lung}\times G\times C_{plasma}^{liver}\times f_{u,p}+Q_{plasma}^{lung}\times G\times C_{endo}^{lung},$$Blood cells
(11)$$V_{BC}^{lung}\times \frac{dC_{BC}^{lung}}{dt}=Q_{BC}^{lung}\times C_{BC}-(Q_{BC}^{heart}+Q_{BC}^{kidney}+Q_{BC}^{brain}+Q_{BC}^{muscle}+Q_{BC}^{bone}+Q_{BC}^{thymus}+Q_{BC}^{skin}+Q_{BC}^{fat}+\;Q_{BC}^{liver}+Q_{BC}^{SI}+Q_{BC}^{LI}+Q_{BC}^{spleen}+Q_{BC}^{pancreas}+Q_{BC}^{other})\times C_{BC}^{lung}+PS_{BC}\times C_{plasma}^{lung}\times f_{u,p}-PS_{BC}\times C_{BC}^{lung}\times \frac{f_{u,p}}{K_{P,BC}},$$Endothelial cell
(12)$$V_{endo}^{lung}\times \frac{dC_{endo}^{lung}}{dt}=Q_{plasma}^{lung}\times G\times f_{u,p}\times C_{plasma}^{lung}-2\times Q_{plasma}^{lung}\times G\times C_{endo}^{lung}+Q_{plasma}^{lung}\times G\times C_{IS}^{lung},$$Interstitial space
(13)$$V_{IS}^{lung}\times \frac{dC_{IS}^{lung}}{dt}=Q_{plasma}^{lung}\times G\times C_{endo}^{lung}-Q_{plasma}^{lung}\times G\times C_{IS}^{lung}-PS_{lung}\times C_{IS}^{lung}+PS_{lung}\times C_{cellular}^{lung}\times \frac{f_{u,p}}{K_{P,lung}},$$Cellular space
(14)$$V_{cellular}^{lung}\times \frac{dC_{cellular}^{lung}}{dt}=PS_{lung}\times C_{IS}^{lung}-PS_{lung}\times C_{cellular}^{lung}\times \frac{f_{u,p}}{K_{P,lung}},$$Typical tissue compartmentsPlasma
(15)$$V_{plasma}^{i}\times \frac{dC_{plasma}^{i}}{dt}=Q_{plasma}^{i}\times C_{plasma}^{lung}-Q_{plasma}^{i}\times C_{plasma}^{i}+PS_{BC}\times C_{BC}^{i}\times \frac{f_{u,p}}{K_{P,BC}}-PS_{BC}\times C_{plasma}^{i}\times f_{u,p}-\;Q_{plasma}^{i}\times G\times C_{plasma}^{i}\times f_{u,p}+Q_{plasma}^{i}\times G\times C_{endo}^{i},$$Blood cells
(16)$$V_{BC}^{i}\times \frac{dC_{BC}^{i}}{dt}=Q_{BC}^{i}\times C_{BC}^{lung}-Q_{BC}^{i}\times C_{BC}^{i}+PS_{BC}\times C_{plasma}^{i}\times f_{u,p}-PS_{BC}\times C_{BC}^{i}\times \frac{f_{u,p}}{K_{P,BC}},$$Endothelial cell
(17)$$V_{endo}^{i}\times \frac{dC_{endo}^{i}}{dt}=Q_{plasma}^{i}\times G\times f_{u,p}\times C_{plasma}^{i}-2\times Q_{plasma}^{i}\times G\times C_{endo}^{i}+Q_{plasma}^{i}\times G\times C_{IS}^{i},$$Interstitial space
(18)$$V_{IS}^{i}\times \frac{dC_{IS}^{i}}{dt}=Q_{plasma}^{i}\times G\times C_{endo}^{i}-Q_{plasma}^{i}\times G\times C_{IS}^{i}-PS_{i}\times C_{IS}^{i}+PS_{i}\times C_{cellular}^{i}\times \frac{f_{u,p}}{K_{P,i}},$$Cellular space
(19)$$V_{cellular}^{i}\times \frac{dC_{cellular}^{tissiue}}{dt}=PS_{i}\times C_{IS}^{i}-PS_{i}\times C_{cellular}^{i}\times \frac{f_{u,p}}{K_{P,i}},$$Tumor CompartmentTumor extracellular
(20)$$\frac{dC_{extra}^{tumor}}{dt}=\left(\frac{2\times P\times R_{cap}}{R_{krough}^{2}}\right)\times \left(C^{plasma}-\frac{C_{extra}^{tumor}}{\varepsilon }\right)+\frac{6\times D}{R_{tumor}^{2}}\times \left(C^{plasma}-\frac{C_{extra}^{tumor}}{\varepsilon }\right)-k_{in}\times C_{extra}^{tumor}+k_{out}\times C_{free,intra}^{tumor},$$Free drug in tumor intracellular
(21)$$\frac{dC_{free,intra}^{tumor}}{dt}=k_{in}\times C_{extra}^{tumor}-k_{out}\times C_{free,intra}^{tumor}-k_{on}^{tub}\times C_{free,intra}^{tumor}\times \left(C_{tubulin}-C_{bound,intra}^{tumor}\right)+\;k_{off}\times C_{bound,intra}^{tumor},$$Bound drug in tumor intracellular
(22)$$\frac{dC_{bound,intra}^{tumor}}{dt}=k_{on}\times C_{free,intra}^{tumor}\times \left(C_{tubulin}-C_{bound,intra}^{tumor}\right)-k_{off}\times C_{bound,intra}^{tumor},$$


#### 2.6.3. Model Fitting and Parameter Estimation

Physiological parameters for mice were obtained from our previously published PBPK model [22]. Values and the sources for tumor-associated parameters are listed in Table 1. Radius of the tumor blood capillary ($R_{cap}$), average distance between two capillaries ($R_{krough}$), tumor void volume for MMAE (ε), P, D, and efflux rate of MMAE from the cells ($k_{out}$) were obtained from the literature [17,18,19]. The equilibrium dissociation rate ($K_{D}$) of MMAE for free tubulin was reported to be 291 nM [20], and the dissociation rate constant ($k_{off}$) of MMAE to tubulin was assumed to be 0.545 (1/h) [16], and thus, the calculated association rate constant ($k_{on}$) of MMAE for tubulin was 0.00187 (1/nM/h). The estimated parameters included (1) permeability coefficient ($PS_{i})$ in all tissues except the intestines, thymus, and other compartments for which observed data were not available, (2) intrinsic hepatic clearance ($CL_{int}$), and (3) nonspecific uptake rate of MMAE into cancer cells ($k_{in})$. The model was fitted to the data using the maximum likelihood estimation method in ADAPT version 5 (Biomedical Simulations Resource, University of Southern California, Los Angeles, CA, USA), using the variance model shown in Equation (23).
(23)$$V_{i}={\left(\sigma_{1}+\sigma_{2}\cdot Y_{i}\right)}^{2},$$

Above, $V_{i}$ represents the variance of the *i*th data point, $Y_{i}$ is the *i*th model prediction, and $\sigma_{1}$ and $\sigma_{2}$ are variance model parameters. The final model performance was evaluated based on observed versus predicted plots, Akaike information criterion, visual inspection of observed versus predicted plots, and CV% of the parameter estimates. For quantitative evaluation of model performance, the percent predictive error (%PE) for plasma and all tissues were calculated using Equation (24).
(24)$$\%PE=\frac{AUC_{0-t,\;\;pred}-AUC_{0-t,\;obs}}{AUC_{0-t,\;obs}}\times 100\%$$

Above, $AUC_{0-t,\;pred}$ is the $AUC_{0-t}$ of the model-predicted PK profiles and $AUC_{0-t,obs}$ is the $AUC_{0-t}$ of the observed PK profiles. 

## 3. Results

Figure 2 shows measured PK profiles of MMAE in plasma, tissues, and tumors. MMAE plasma concentration dropped rapidly, with only 0.3% of the injected dose remaining after 5 min. Plasma concentration was below the limit of quantification (BLQ) after 12 h. Prolonged MMAE concentrations in tissues and tumor were observed when compared to plasma. MMAE concentration in the tumor remained steady, and the concentration decreased by only 50% from 1 to 168 h. MMAE concentrations in all tissues except the liver were quantifiable at 24 h. MMAE concentration in the liver, which was the primary elimination tissue for the drug, was quantifiable only up to 6 h. Noncompartmental analysis (NCA) showed that the plasma AUC of MMAE from 0 to 12 h was 54.3 ng·h/mL, and the AUC from 0 to infinite was 54.5 ng·h/mL, which indicates that the majority of the systemic exposure of MMAE was limited to 12 h. MMAE demonstrated rapid systemic CL (60 mL/h), a short half-life (2.5 h), and large volume of distribution ($V_{ss}$ = 42 mL), which suggests an extensive tissue distribution of MMAE. 

Table 2 shows $AUC_{0-t}$ values for plasma, tissues, and tumor, and $K_{p}$ values before and after adjusting for the fractional cellular space volume. $K_{p}$ values were 1.2-fold (muscle and fat) to 1.8-fold (lung) higher after adjustments. The $K_{p}$ for red blood cells was 5.5, indicating that the MMAE distribution is relatively high in red blood cells. MMAE rapidly and extensively distributed into tissues and was retained locally, which led to $K_{p}$ values >1 in all tissues, ranging from 1.25 to 35.3, except for in the brain. The apparent $K_{p}$ in the liver was adjusted for $CL_{int}$, since it is the eliminating tissue. After accounting for $CL_{int}$, $K_{P}$ in the liver with and without adjustments for the fractional cellular volume was 16.1 and 25.3, respectively. Based on the tissue concentration–time profiles and $K_{p}$ values, MMAE tissue distribution kinetics could be classified into three groups. First, in highly perfused tissues, including lung, kidney, heart, liver, and spleen, rapid and extensive MMAE distribution was observed with tissue-to-plasma AUC ratios > 20. Second, in poorly perfused tissues including fat, pancreas, skin, bone, and muscle, $K_{p}$ ranged from 1.3 (muscle) to 2.4 (fat). Third, MMAE distribution was limited in the brain, with brain exposure only being half of the systemic exposure. Of note, MMAE remained steadily in the tumor for 168 h, while only a small amount of drug was detectable in the plasma at later time points. The overall exposure of MMAE in MDA-MB-468 tumors was eight times higher than plasma exposure.

Figure 3 shows observed and PBPK model fitted MMAE PK profiles in plasma, tissues, and tumors. The PBPK model well captured the PK in plasma, tissues, and tumor simultaneously. Table 3 summarizes the PBPK model parameters estimated using the data. $CL_{int}$ was estimated with good precision (%CV = 8.92), and the optimized value was 137 mL/h. The estimated $CL_{int}$ value from the PBPK model fitting was comparable to values extrapolated from in vitro $CL_{int}$ ($CL_{int,\;vitro})\;$ values, which were calculated using MMAE metabolism studies in human liver hepatocyte (3 µL/min/10^6^ cells) or human liver microsome (24 µL/min/mg) [21]. The in vivo $CL_{int}$ values calculated using hepatocytes and microsomes were 101 and 187 mL/h, respectively. The PBPK model estimated $CL_{int}$ value was also similar to the $CL_{int}$ value derived from our NCA analysis of in vivo PK data, where the calculated value was 150 mL/h. Please refer to the Discussion section for a detailed derivation of $CL_{int}$. The rate of non-specific uptake of MMAE into tumor cells was estimated with good precision (%CV = 19.3), and the optimized value was 0.182 L/h. permeability-surface area coefficient (PS) values were estimated with good precision (%CV < 40) in blood cells and in most tissues, with slightly lower confidence in the estimation of PS value for the liver (%CV < 50). The estimated PS in blood cells was much lower than the blood flow rate, indicating permeability-limited drug transfer between blood cells and plasma. Similarly, the estimated PS in each tissue was much lower than the tissue blood flow rate, which confirms that the distribution of MMAE was slow and permeability-limited between interstitial and cellular spaces.

Table 4 summarizes a quantitative comparison of observed and PBPK model predicted PK profiles of MMAE in the form of %PE. The model best characterized lung, heart, kidney, and skin data with %PE < 0.6%. The model also well captured the data in plasma, blood cell, spleen, brain, fat, bone, and tumor with %PE < 10%, which was followed by muscle and pancreas with %PE about 15%. The model slightly underpredicted liver concentrations (%PE = −38.7).

## 4. Discussion

Whole-body PK of MMAE is essential for understanding the toxicity of MMAE-conjugated ADCs. However, no studies have comprehensively quantified MMAE disposition in vivo. Here, we have presented the first-ever whole-body biodistribution study of MMAE in tumor-bearing mice, where MMAE concentrations were quantified in plasma, 11 tissues (e.g., heart, liver, lung, spleen, kidney, pancreas, fat, brain, skin, muscle, and bone), and tumor. We have established a PBPK model and incorporated a cell-level tumor PK model to simultaneously characterize MMAE disposition in systemic circulation and at the site-of-action. This MMAE PBPK model can serve as a stepping stone to develop a platform PBPK model for MMAE-conjugated ADCs, which can be used to facilitate preclinical-to-clinical translation and better understand the safety and efficacy of ADCs in the clinic.

The PK profile of MMAE in plasma dropped rapidly, while concentrations in tissues were retained for a prolonged period of time. This could be because of the binding of MMAE to its intracellular target, tubulin. Conventional moment analysis of PK data showed fast CL and large volume of distribution for MMAE, which corresponds with the observed tissue concentration profiles. Both PK profiles and NCA indicated that MMAE distributed extensively into the tissues, and it may have been retained within the tissue cells. Since plasma and tissue concentration profiles were not parallel, plasma exposure alone may not be sufficient to serve as a driver for MMAE-induced toxicity. The half-life of MMAE in our study was slightly shorter than the reported value (2.5 h vs. 5.7 h), and as expected, it was considerably shorter than the half-life reported after the administration of MMAE-conjugated ADC (2.5–3 days), which is confounded by the formation-rate limited kinetics [4,16,23]. 

Our results regarding the blood cell distribution of MMAE were not consistent with the literature. The MMAE blood-to-plasma ratio is reported to be 2 [4] and 1.53–8.65 [8] in different studies, following free MMAE administration. Whereas, a lower blood-to-plasma ratio has been observed when MMAE-conjugated ADCs were administrated [4]. Our results showed that the MMAE blood-to plasma ratio was about 6 at the early time-points (<30 min), and the ratio increased in the later time point (e.g., ≈20 at 6 h). 

Concentration data were not obtained for gastrointestinal tracts and thymus, and thus, an assumption was made when calculating $K_{P}$ values for these tissues based on the result of a quantitative whole-body autoradiography study of trastuzumab-vc-MMAE in rats [7]. The study observed persistent MMAE radioactivity in tissues with rapidly dividing cells such as GI epithelia, bone marrow, spleen, and thymus. Accordingly, apparent $K_{P}$ in GI tracts and thymus were assumed to be the same as in spleen, and they were adjusted for the fractional cellular volume of each tissue. $K_{P}$ values were >1 in all tissues, except the brain, which indicates that MMAE exposure was higher in most tissues compared to the plasma. Extensive distribution of MMAE was observed in highly perfused tissues (i.e., lung > spleen > liver > heart), where exposure was >15-fold higher than the plasma exposure. Moderate distribution of MMAE was observed in fat, pancreas, skin, bone, and muscle, where exposure was about 2-fold higher than plasma exposure. These findings correspond well with the results from a radioactivity study [8] and a quantitative whole-body autoradiography study [7], which both reported that radioactivity was relatively high in highly perfused tissues on day 1 post-dose. Brain $K_{P}$ was relatively low, and the exposure of brain was about half of the systemic exposure. It is reported that MMAE is a substrate of P-glycoprotein transporters [5], which may be involved in MMAE efflux out of the brain. As such, the differences in MMAE tissue distribution can be explained by the differences in blood perfusion rate, tissue cell membrane penetration rate, or the involvement of transporters.

The PBPK model was able to well characterize the PK of MMAE in plasma, tissues, and tumor. The model predicted and observed concentrations for lung, heart, skin, and kidney were very similar (%PE < 0.6%). The %PE for the rest of the non-eliminating tissues were <16%. Furthermore, the cell-level tumor disposition model was incorporated into the PBPK model, and the model successfully captured MMAE PK in tumors, which showed prolonged retention up to 168 h. The MMAE PBPK model proposed here accounts for drug transfer kinetics between different physiological sub-compartments and accounts for important factors in tissue distribution. These factors include perfusion-limited drug transfer between plasma and extracellular space, permeability-limited drug transfer across the cell membrane, red blood cell distribution, protein binding, drug–target interaction, and cellular partition. Furthermore, since tissue-level compartments were kept the same as in our published platform PBPK model for antibody [22], one can further combine this MMAE PBPK model with the antibody PBPK model to build a platform PBPK model of MMAE-conjugated ADCs [13].

It is reported that the elimination (metabolism and excretion) of MMAE is mainly through CYP3A4-mediated metabolism, biliary/fecal excretion, or urinary excretion [4]. About 80% of MMAE excretes via feces, and only 6% excretes via urine. Therefore, we assumed that MMAE was eliminated only via metabolism and biliary/fecal excretion from the liver. The $CL_{int}$ estimated by the PBPK model was comparable to the values calculated using in vitro and in vivo data. For the derivation of $CL_{int}$ based on in vitro data, the $CL_{int,\;vitro}$ were estimated using human hepatocyte (3 µL/min/10^6^ cells) and human liver microsome (24 µL/min/mg) [21]. An in vitro–in vivo extrapolation was applied to calculate in vivo metabolic CL ($CL_{in\;vivo}$) of MMAE, using the Equation (25).
(25)$$CL_{in\;vivo}=CL_{int,\;in\;vitro}\cdot scaling\;factors\cdot mice\;liver\;weight$$

The scaling factors used were 135 ± 10 (10^6^ cells/g) for hepatocytes [24] and 41.9 ± 24.5 (mg/g) [25] for microsome studies. Metabolic CL and non-metabolic (mostly biliary) CL were assumed to account for 40% and 60% of total CL. This assumption is based on a human mass balance study [4] and a bile-duct cannulated rat study, which reported that ≈60% of MMAE is excreted unchanged in the bile [9]. As such, the $CL_{int}$ extrapolated from in vitro study was 109–126 mL/min (hepatocyte) and 121–462 mL/min (microsome). The $CL_{int}$ approximated from our in vivo data was 146 mL/min using the conventional moment analysis and assuming a well-stirred model, low extraction of the drug, and concentration-independent CL.

Plasma protein binding plays a vital role in tissue distribution of small molecule drugs. It is reported that MMAE plasma protein binding is species-dependent, with higher levels in rats and humans (67.9–82.2%) compared to mice and monkeys (17.1–28.5%) [4]. Based on the reported mice data, $f_{u,p}$ was assumed to be 0.2, and it was used as a concentration-independent parameter in the PBPK model. Of note, species-dependence of fraction unbound should be considered when further scaling up the current MMAE PBPK model to different species. Based on free hormone hypothesis [26], it was assumed that only free MMAE could transfer across the cell membrane, and thus, the fraction of free drug in each tissue was considered in the PBPK model. By assuming linear and time-invariant conditions, the fraction of unbound drug in each tissue and blood cell was calculated (Equation (2)) and included in the PBPK model.

PS describes the permeability-limited transfer of the drug between plasma/blood cell and interstitial/cellular space. Compared to the regional blood flow rate, the relatively low PS values indicated that it was necessary to account for the permeability property in the PBPK model. Furthermore, PS values can be scaled up using an allometric equation with the exponent of 0.67 [27,28], which is regularly used for body surface area scale-up. The fixed value of 0.67 exponent for PS is typically used under the assumption that the permeability of the tissue cellular membrane and organ structure is similar across different species [28]. On the other hand, the direct scale-up of CL estimated from the PBPK model should be critically evaluated. In our PBPK model, which was established based on mice data, the liver was the only clearance organ for MMAE. Whereas, kidney excretion is reported to be higher in humans (i.e., 23–41%) [4]. Therefore, extrahepatic clearance and species differences should be considered when scaling-up the MMAE PBPK model to higher species.

In the clinic, the incidences of hematological and neurological toxicities are the highest for MMAE among all other commonly used payloads. Three major toxicities, including peripheral neuropathy, anemia, and neutropenia, have been consistently reported in patients treated with MMAE-conjugated ADCs [6]. Surprisingly, peripheral neuropathy, which was observed in up to 50% of patients receiving brentuximab vedotin, was not predicted based on preclinical toxicology studies in rats and monkeys [29]. One possible explanation was that only plasma PK but not tissue PK of MMAE was measured [30,31], which emphasizes that solely using systemic PK may not be able to establish a translatable exposure–toxicity relationship for ADCs. Our study reported that the exposure of MMAE in tissues and red blood cell is relatively higher and sustained for longer period of time compared to plasma exposure, which could lead to a high prevalence of peripheral neuropathy (i.e., numbness and tingling in the extremities) via the arrest of microtubule networks, disruption of the axonal transport, and degeneration of peripheral nerve terminals [12]. Another explanation for the toxicity could be that a high MMAE concentration in blood cells causes an apoptosis of red blood cells, resulting in decreased blood flow and lack of nutrients transported to the peripheral nerve, which gradually prompts the degeneration. In addition, bone marrow toxicity such as anemia and neutropenia is expected from MMAE, since it is a tubulin inhibitor that targets rapidly proliferating cells [4]. Our calculated $K_{P}$ for bone was >1 and MMAE concentration in bone retained over time, both of which supports the high frequency of bone marrow toxicity in the clinic.

In a phase II study of brentuximab vedotin, the probability of objective response rate (ORR) was reported to decrease with increased MMAE trough concentration [32,33]. In addition, there is a report suggesting that the side effect of ADC increased with decreasing MMAE trough concentrations [32,33]. While no clear explanation has been given for these observations, they suggest that plasma concentrations of free MMAE may not represent the toxicity/efficacy of MMAE ADCs, and the tissue distribution of MMAE needs to be accounted for. While our data highlight the importance of measuring/predicting the tissue PK of free payloads, it is important to note that in the clinic, the measurements are mainly done in the plasma, and plasma ADC PK can still serve as a viable driver to establish exposure–response relationships for MMAE-conjugated ADCs. Additionally, considering the formation-rate limited nature of free MMAE exposure after ADC administration and the different fraction unbound of MMAE among different species, our observed PK of free MMAE in mice may not be directly relevant for establishing toxicodynamic relationships for MMAE-conjugated ADCs in the clinic.

The PBPK model of MMAE presented here can further be used to assess drug–drug interactions of MMAE containing ADCs. In vivo and in vitro studies indicate that MMAE is a substrate of CYP3A and P-glycoprotein [34]. As such, the polymorphism of CYP3A and P-glycoprotein may affect the PK of MMAE, and consequently the efficacy and toxicity of MMAE-conjugated ADCs. As such, the MMAE PBPK model can serve as a tool to explore drug interactions of ADCs.

There are some limitations of our study. First, only one dose level of MMAE was used to build the PBPK model. However, the current 0.1 mg/kg MMAE dose can be high enough to saturate the target, and thus, linear PK can be assumed. This assumption was based on the results of toxicokinetic studies in rats, which reported that single-dose no-observed-adverse-effect level (NOAEL) was 0.01 mg/kg and >0.116 mg/kg induced mortality [4]. As more data at lower dose levels become available, the PBPK model can be further refined to elaborate the MMAE–tubulin interaction. Second, the PK profile of free MMAE in tumor presented in this study and predicted by the PBPK model may not represent the exposure of MMAE observed after MMAE-conjugated ADC administration. An ADC utilizes the monoclonal antibody component to deliver cytotoxic drugs in the antigen-expressing tumor. After the target binding, an ADC undergoes internalization and payload release, followed by payload accumulation at the site of action. On the other hand, MMAE PK in the tumor following intravenous administration of the MMAE is independent of ADC–target interaction, and it is mainly driven by the payload’s ability to enter and remain in the tumor tissue and cancer cells. As such, payload-specific parameters (e.g., permeability, diffusivity, etc.) and tumor-specific parameters (e.g., vascular density, tumor size, etc.) together contribute to the MMAE PK profiles reported by us. Consequently, the rate and extent of free MMAE exposure in tumor presented in this study could be different than the one observed following MMAE-conjugated ADC administration. Additionally, since the systemic exposure of an ADC is relatively longer compared to free MMAE, the total MMAE exposure observed following ADC administration is dominated by conjugated MMAE, and the free MMAE contributes minimally (i.e., <20% based on our in-house data) to tissue total MMAE exposure. Therefore, the disposition of free MMAE in tissues observed following ADC administration could be different than the one presented in this study. 

## 5. Conclusions

Here, we have investigated the whole-body PK of MMAE at a 0.1 mg/kg single dose in tumor-bearing mice. We observed that while MMAE is rapidly eliminated from the systemic circulation (i.e., plasma), it shows prolonged retention in tissues, tumor, and blood cells. We have also developed a PBPK model for MMAE, which accounts for perfusion/permeability-limited transfer of the drug to the tissues, blood cell distribution of the drug, tissue retention of the drug, and protein binding. The model was able to characterize the PK of MMAE in plasma, tissue, and tumor reasonably well, and the model parameters were estimated with good confidence. The MMAE PBPK model presented here can serve as the first step to building a platform PBPK model for MMAE containing ADCs.

## Figures and Tables

**Figure 1 jcm-10-01332-f001:**
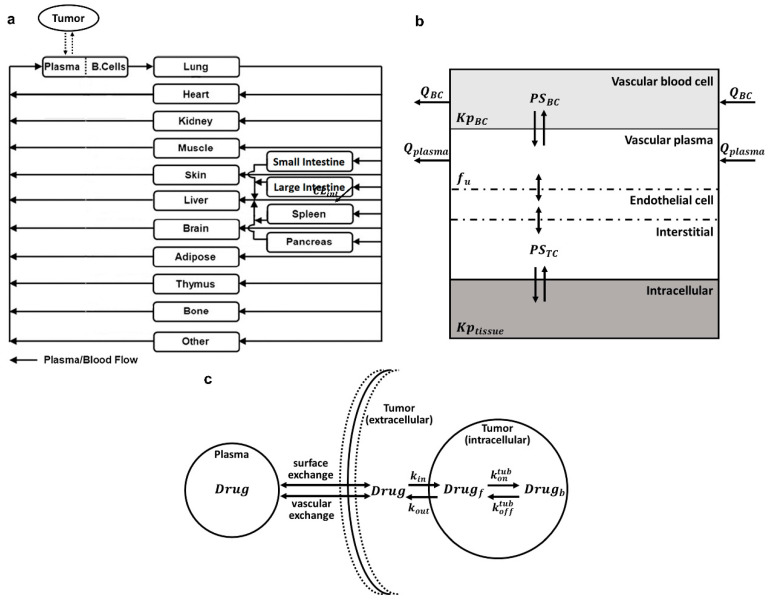
Structure of monomethyl auristatin E (MMAE) physiological-based pharmacokinetic (PBPK) model. (**a**) Structure of the whole-body PBPK model for MMAE. All tissue compartments are connected in an anatomical manner with blood flow indicated by the solid arrows. (**b**) Structure of the tissue level PBPK model for MMAE. Each tissue compartment is divided into the vascular, endothelial cell, interstitial, and cellular sub-compartments. The vascular sub-compartment is further divided into plasma and blood cells. For a detailed description of the symbols and drug disposition processes, please refer to the model structure section in the method section. (**c**) Schematics of the cell-level tumor disposition model for MMAE. For a detailed description of the symbols and drug disposition process, please refer to Table 1 and Model structure section in the Materials and Methods.

**Figure 2 jcm-10-01332-f002:**
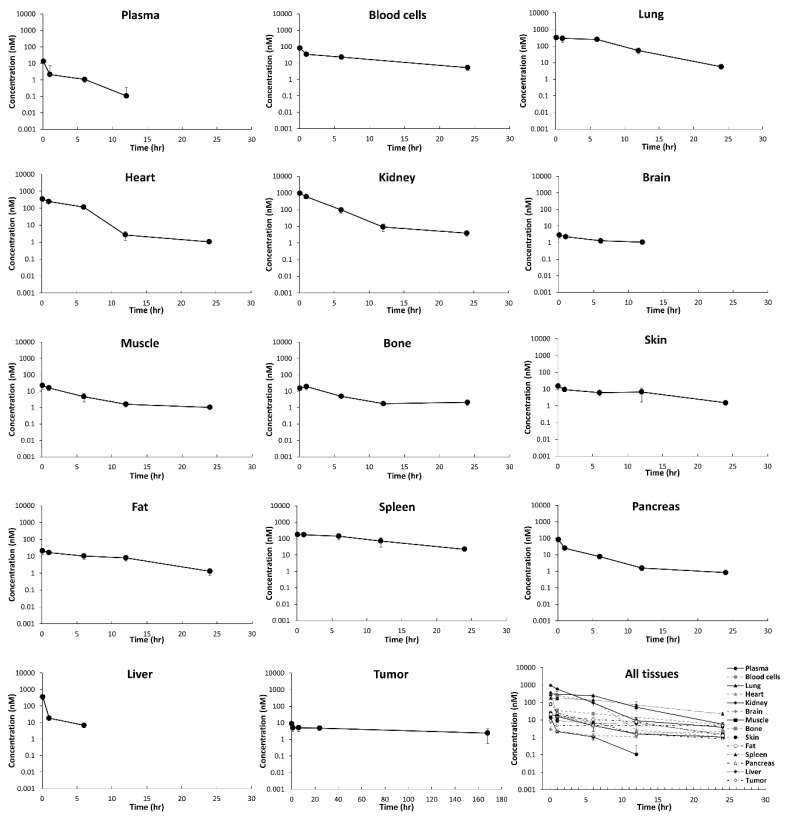
Observed whole-body pharmacokinetics (PK) of MMAE in mice after intravenous administration of 0.1 mg/kg MMAE dose. The figure displays the mean (SD) observed concentration (black dots) in plasma, tissues, and tumor. All the PK profiles (truncated to 24 h) are superimposed in the last panel.

**Figure 3 jcm-10-01332-f003:**
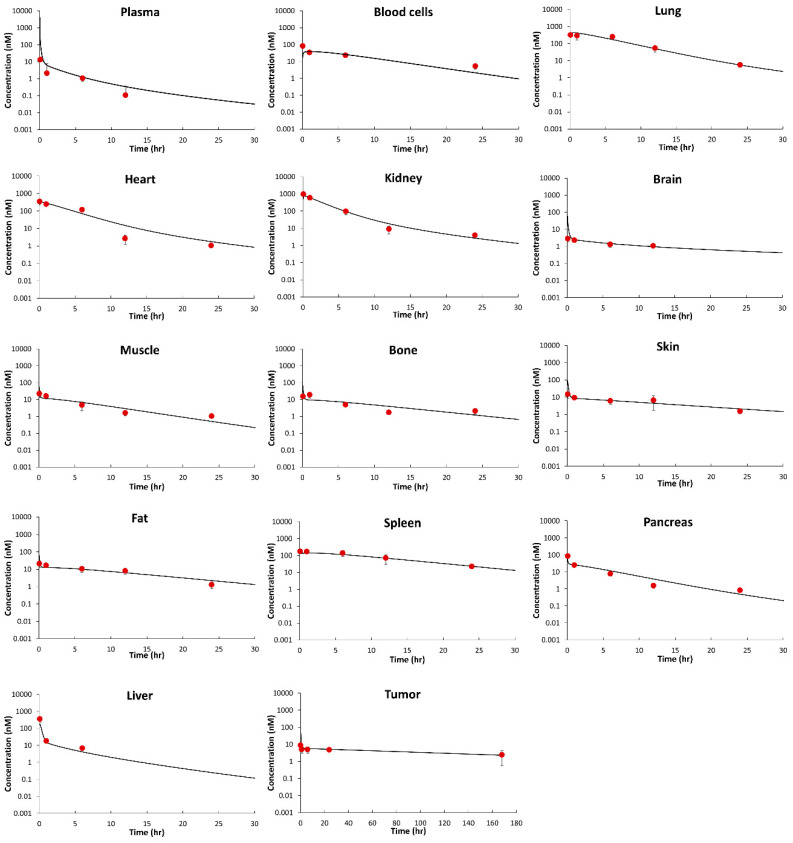
Comparison of PBPK model fitted and observed PK profiles of MMAE. The figure displays observed (dots) and model predicted (solid lines) plasma, tissues, and tumor concentration vs. time profiles of MMAE in mice after intravenous administration of 0.1 mg/kg dose.

**Table 1 jcm-10-01332-t001:** A glossary of literature-derived and estimated parameters used for the MMAE PBPK model.

Tissue	Definition	Value	Unit	Source
$f_{u,p}$	Fraction unbound in plasma	0.8	-	[4]
$R_{cap}$	Radius of tumor blood capillary	0.8	cm	[17,18,19]
$R_{krough}$	Average distance between two capillaries	7.5	cm	[17,18,19]
$R_{tumor}$	Tumor radius	0.7	cm	Obtained from mice
P	Permeability rate across the blood vessels	87.5	cm/h	[17,18,19]
D	Diffusion rate across the blood vessels	0.0104	cm^2^/h	[17,18,19]
ε	Tumor void volume	0.44	-	[17,18,19]
$k_{on}$	Second-order association rate constant between cytoplasmic MMAE and intracellular tubulin protein	0.00187	1/nM/h	Based on $K_{D}$ from [20]
$k_{off}$	First-order dissociation rate constant between MMAE–tubulin complex	0.545	1/h	Assumed from [16]
$k_{in}$	First-order influx rate of MMAE from extracellular to intracellular space	0.185 (%CV = 19.3)	1/h	Estimated
$k_{out}$	First-order efflux rate of MMAE from intracellular to extracellular space	0.046	1/h	[17,18,19]
$C_{tubulin}$	Total tubulin concentration	2000	nM	[21]
G	Factor multiplied by tissue plasma flow to make drug distribution instantaneous	1000	-	Assumed

**Table 2 jcm-10-01332-t002:** *AUC*_0−*t*_ and partition coefficient ($K_{P})$ of MMAE.

Tissue	${\mathrm{AUC}}_{0-t}\;{(h\cdot \mathrm{nM})\;}^{1}$	$K_{P}{\;}^{2}$	$K_{P}^{adj}{\;}^{3}$
Plasma	75.4	-	-
Blood	414	5.46	5.46
Lung	2679	35.3	64.9
Heart	1356	17.9	22.8
Kidney	2396	31.6	42.4
Brain	31.2	0.411	0.530
Muscle	94.6	1.25	1.51
Bone	110	1.45	1.89
Skin	130	1.71	2.87
Fat	184	2.43	3.01
Spleen	2056	27.1	47.2
Pancreas	160	2.11	2.93
Liver	176	2.42	3.80
Tumor	619	8.21	-

^1^*AUC*_0−*t*_ were calculated using observed MMAE concentration. ^2^ Tissue-to-plasma partition coefficients ($K_{P}$) were calculated as ratios of tissue *AUC*_0−*t*_ and plasma *AUC*_0−*t*_. ^3^
$K_{P}$ values were adjusted for the percentage of cellular volume in total tissue volume. - not applicable.

**Table 3 jcm-10-01332-t003:** PBPK model estimated parameter values.

Parameters	Estimated (CV%)
$PS_{blood},\mathrm{mL}/h$	0.105 (12.6)
$PS_{lung},\mathrm{mL}/h$	2.47 (13.2)
$PS_{heart},\mathrm{mL}/h$	1.47 (17.1)
$PS_{kidney},\mathrm{mL}/h$	14.2 (20.2)
$PS_{brain},\mathrm{mL}/h$	0.00825 (40.4)
$PS_{muscle},\mathrm{mL}/h$	3.16 (21.1)
$PS_{bone},\mathrm{mL}/h$	0.568 (20.9)
$PS_{skin},\mathrm{mL}/h$	0.681 (30.6)
$PS_{fat},\mathrm{mL}/h$	0.588 (23.5)
$PS_{spleen},\mathrm{mL}/h$	0.457 (18.5)
$PS_{pancreas},\mathrm{mL}/h$	0.0657 (18.7)
$PS_{liver},\mathrm{mL}/h$	49.2 (48.4)
$CL_{int},\mathrm{mL}/h$	137 (8.92)

PS, permeability-surface area coefficient; $CL_{int}$, liver intrinsic clearance.

**Table 4 jcm-10-01332-t004:** Percentage predictive errors for the quantitative comparison of observed and PBPK model predicted MMAE PK profiles.

Tissue	% PE
Plasma	8.21
Blood	8.16
Lung	0.0161
Heart	0.534
Kidney	0.602
Brain	9.63
Muscle	12.6
Bone	6.53
Skin	0.504
Fat	6.83
Spleen	9.85
Pancreas	16.1
Liver	38.7
Tumor	7.37

Percent predictive error (%PE) were calculated as: $\left|AUC_{predicted}-AUC_{observed}\right|/AUC_{observed}\cdot 100\%$.

## Data Availability

Data generated is included within the manuscript.

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
