# Peer review of "Whole-Body Pharmacokinetics and Physiologically Based Pharmacokinetic Model for Monomethyl Auristatin E (MMAE)"

_jcm, 2021, doi:10.3390/jcm10061332_

Round 1

Reviewer 1 Report

Interesting paper/approach.  PBPK model well characterized and predicts data well. 

Points of consideration - 

Overall - I think should more should be discussed on the limitations of these data with regards to free MMAE, Fu and translation to clinic.  I acknowledge that free MMAE, once released, would have similar PK properties, but this avoids the whole discussion of the ADC, and internalization, linker stability, etc.

1) Recommend elaboration on the role of the ADC in the process of uptake/internalization into tissues and especially tumor in the discussion. Need to relate the free MMAE administered IV and uptake into tumor vs an ADC which is ultimately designed to drive uptake in the tumor.  Your concentrations of free would have assumed all MMAE was released by the ADC and uptake into the tumor.  This would alter the components of exposure (i.e. Cmax vs AUC) for MMAE vs ADC.  

2) More discussion is needed on the role of the stability of an ADC (i.e. linker drug).  Similar to #1 - the assumption that these levels of MMAE would have similar retention as an ADC in tissue (uptake via pinocytosis) vs free MMAE uptake.

3) Lines 477-483 - while these examples are included/explored as reasons for limitations using MMAE, most would agree that obtainable data from measuring the ADC has better plasma exposure-response relationship (Cf. to MMAE).   It seems to imply that these data in the manuscript might add some value to the understanding, but should be TD of MMAE released from the ADC in these situations, and not necessarily these data of free MMAE in the mouse (with lowered Fu compared to humans would be more relevant. 

Line 57 - wrong citation included.

Reviewer 2 Report

In this work, Hsuan-Ping Chang et al describe a physiological-based pharmacokinetics to characterize whole-body PK of MMAE. The manuscript is well written with the introduction and discussion constructed in an appropriate manner, and also the contribution of this novel PBPK model can be considered relevant. Only minor revisions are requested before publication:

  • The evaluation of free MMAE PK in tumor bearing mice has the characteristic for which toxin retainment in tumor tissue is independent from ADC target, but dependent only on the tissue and cell features of the tumor tissue, so quantity and retainment time of MMAE in tumor could be change in the case of use of conjugated-MMAE, which has the important ability to accumulate free toxin in antigen-positive tumor. Please add a sentence about this limit of the model used in absence of comparison with MMAE-conjugated ADC.
  • Reference number in line 57 is wrong. Please correct.
